# Aortic Valve Stenosis and Mitochondrial Dysfunctions: Clinical and Molecular Perspectives

**DOI:** 10.3390/ijms21144899

**Published:** 2020-07-11

**Authors:** Gaia Pedriali, Giampaolo Morciano, Simone Patergnani, Paolo Cimaglia, Cristina Morelli, Elisa Mikus, Roberto Ferrari, Vincenzo Gasbarro, Carlotta Giorgi, Mariusz R. Wieckowski, Paolo Pinton

**Affiliations:** 1Maria Cecilia Hospital, GVM Care & Research, Cotignola, 48033 Ravenna, Italy; pdrgai@unife.it (G.P.); mrcgpl@unife.it (G.M.); simone.patergnani@unife.it (S.P.); fri@unife.it (R.F.); 2Department of Medical Sciences, Laboratory for Technologies of Advanced Therapies (LTTA), University of Ferrara, 44121 Ferrara, Italy; vincenzo.gasbarro@unife.it (V.G.); grgclt@unife.it (C.G.); 3Cardiovascular Department, Maria Cecilia Hospital, GVM Care & Research, Cotignola, 48033 Ravenna, Italy; paolocimaglia88@gmail.com (P.C.); elisamikus@yahoo.it (E.M.); 4Cardiology Unit, Azienda Ospedaliero Universitaria di Ferrara, 44121 Ferrara, Italy; morelli.cri@gmail.com; 5Laboratory of Mitochondrial Biology and Metabolism, Nencki Institute of Experimental Biology, Pasteur 3, 02-093 Warsaw, Poland; m.wieckowski@nencki.edu.pl

**Keywords:** Aortic stenosis, mitochondria, inflammation, cardiovascular disease, autophagy

## Abstract

Calcific aortic stenosis is a disorder that impacts the physiology of heart valves. Fibrocalcific events progress in conjunction with thickening of the valve leaflets. Over the years, these events promote stenosis and obstruction of blood flow. Known and common risk factors are congenital defects, aging and metabolic syndromes linked to high plasma levels of lipoproteins. Inflammation and oxidative stress are the main molecular mediators of the evolution of aortic stenosis in patients and these mediators regulate both the degradation and remodeling processes. Mitochondrial dysfunction and dysregulation of autophagy also contribute to the disease. A better understanding of these cellular impairments might help to develop new ways to treat patients since, at the moment, there is no effective medical treatment to diminish neither the advancement of valve stenosis nor the left ventricular function impairments, and the current approaches are surgical treatment or transcatheter aortic valve replacement with prosthesis.

## 1. Introduction

Aortic stenosis (AS) is the most frequent heart valvular disease in Western countries and is considered an important public health problem, as confirmed by a long-term study in which the impact of this pathology has been evaluated in 185 countries [1]. The data presented revealed a predominance in individuals 70 years and older and a similar prevalence rate of the disease in both men and women.

Heart valves are tissue structures that permit unobstructed, one-way blood flow without regurgitation. Their performance is amazing, maintaining adequate strength and endurance to physically resist to constant and extensive mechanical stress and strain every day. It has been estimated that the aortic valve opens and closes at a rhythm of 60 times/min for >3 billion times during an average lifetime [2].

The typical aortic valve possesses three leaflets, and each one has a trilaminar structure as described by Figure 1 [3].

The cell types described in Figure 1 are involved in physiology and pathobiology; through their complex interactions with the extracellular matrix (ECM), they sense mechanical changes and transform these stimuli into different cellular pathways, such as repairing tissue injury [4].

Valvular endothelial cells (VECs) differ from other vascular endothelial cells; for example, porcine aortic endothelial cells and porcine VECs in response to shear stress activate different gene expression programs [5]. Additionally, VECs are aligned perpendicularly to the orientation of shear stress, whereas other vascular endothelial cells are parallel to the flow [5]. Furthermore, VECs are capable of modulating valvular interstitial cell (VIC) phenotype and matrix synthesis in order to guarantee the integrity of valve tissues; in a 3D valve leaflet model composed of valvular endothelium and interstitial cells, VECs influence VIC proliferation, stimulating their differentiation to a more quiescent phenotype that is characterized by decreased expression of α-smooth muscle actin [5]. Moreover, pig aortic valves have shown a different transcriptional profile in the endothelium on the aortic side versus the ventricular side, identifying distinct endothelial phenotypes among these cells, depending on what the cells are facing [6].

VICs participate in multiple functions, for example, they express matrix metalloproteinases (MMPs) that are implicated in tissue turnover and matrix remodeling; the function of these proteins is balanced by tissue inhibitors of metalloproteinases (TIMPs), whose expression has been revealed also in heart valves [7]. In detergent-treated valve bioscaffolds experiments, VICs, obtained from leaflet microexplants, are able to repopulate the leaflets and to generate a diversified mesenchymal cell population composed of fibroblasts, myofibroblasts, smooth muscle cells (SMCs), and endothelial cells [8]. Normal human aortic valves present few SMCs at the base of the ventricularis, which are characterized by early and late markers of differentiation. In contrast, smooth muscle markers and coactivators of smooth muscle gene expression are not expressed in calcified valves, suggesting the involvement of this type of cell in calcific aortic valve stenosis (CAVS) progression [9].

Cell energy and metabolic activity derive from mitochondrial function throughout body, and mitochondrial dysfunction is considered a therapeutic target for heart diseases [10]. Defects in mitochondrial biogenesis, fission–fusion dynamics and bioenergetics have been observed in several studies performed in humans, animals and in vitro models and are strongly associated with cardiovascular diseases. Often, these effects are caused by genetic mutations in mtDNA, altered expression of mitochondrial proteins or important alterations in metabolic and mitophagic pathways. Mitochondrial dysfunctions may also occur in valvular diseases, as suggested in 2006 by Shinde S. et al. [11], but very little is known about AS.

## 2. Heart Valve Stenosis Epidemiology, Diagnosis and Therapies

Over the years, several studies have been performed to examine the incidence and progression of calcinosis and degeneration of the aortic valve. In particular, while light alcohol consumption is related with a lower risk of AS; cigarette smoking, hypertension, diabetes, renal insufficiency, visceral obesity and metabolic syndrome seem to have an involvement in the development of AS [12]. The pathophysiological mechanism for degeneration of the aortic valve is a dynamic process of inflammation, which determines progressive thickening and the calcification of leaflets. The progressive reduction in leaflet motility and narrowing of the orifice lead to increased afterload which, in turn, causes left ventricular hypertrophy, myocardial fibrosis, systolic and diastolic dysfunction and eventually heart failure [13]. Typically, patients become symptomatic when the aortic valve area is severely reduced (<1 cm^2^), with symptoms such as angina, dyspnea and syncope, usually related to exercise [14]. Echocardiography is the gold standard for the assessment of the anatomy and function of the aortic valve. Patients with CAVS exhibit calcific nodules on the cusps, reduced opening, and a high pressure gradient across the valve. Currently, there is no effective medical treatment to reduce either the progression of valve stenosis or symptoms and mortality; thus, surgical aortic valve replacement (AVR) with mechanical or biological prostheses has been the gold standard treatment for decades. The improvement of transcatheter AVR technologies has permitted a less invasive intervention for elderly patients with severe AS [15]. It is of note that two very recent randomized control trials from independent groups revealed that, in patients with severe AS at low surgical risk, transcatheter AVR achieved better results than surgical intervention on the rate of the composite of death, stroke, or rehospitalization at 1 year [16,17]. Frail patients with contraindications to surgical and transcatheter AVR may only temporarily benefit from palliative balloon valvuloplasty [18].

## 3. Aortic Stenosis Pathobiology, a Molecular Perspective

Degenerative AS manifests as inflammation associated with basement membrane discontinuity and with lipid deposits in the tissue, deeply infiltrated by macrophages and T cells [19]. The degree of inflammatory process is strictly linked to active dynamic transformation of the tissue and correlates with the grade of the pathology [20]. Normal valve structure consists of a well-organized ECM and specific VIC distribution to permit efficient functionality of the heart, but this precise composition is absent in the stenotic bicuspid aortic valves (BAVs) of pediatric patients in whom there is dysregulation of the ECM and impaired VIC organization [21].

At the early stage of aortic calcification, endothelial cells are dysfunctional, causing amplified mechanical stress and diminished shear stress [13], which results in increased cell permeability, adhesion, and proliferation (Figure 2). This process is called aortic sclerosis and is similar to atherosclerosis, which is related to systemic endothelial dysfunction that facilitates lipids diffusion and deposition. Stenotic valve samples revealed an increase in plasma lipoprotein particles deeply modified by oxidative and lipolytic processes [22]. Accumulation of different apolipoproteins (apo) in the aortic valve leads to the progression of the stenotic process and inflammation, as shown by the presence of chronic inflammatory cells, macrophages and T cells close to the lesion [23,24]. Capoulade et al. found an association with elevated lipoprotein a (Lp) and oxidized phospholipid-lipoprotein(b) (OxPL-apoB) levels with rapid AS evolution and the requirement for surgical intervention [25]. These results were confirmed by a multimodality imaging study, in which, increased Lp(a) and OxPL-apoB levels were connected with advanced AS, and in vitro experiments on VICs have demonstrated pro-osteogenic functions of Lp(a) depending on the OxPL concentration [26]. In fact, in vitro administration of Lp(a) on human aortic valvular interstitial cells (HAVICs) induced an osteogenic differentiation, an accumulation of phosphate and calcium content, and increased apoptosis [27]. Autotaxin is a phospholipase enzyme able to interact with Lp(a) in HAVICs and induce inflammation and mineralization of the cells and murine aortic valves [28]. Furthermore, the same group performed a case-control study on patients with CAVS, which presented higher levels of autotaxin associated with higher levels of oxPLs and Lp(a) [29]. In addition, a recent study showed an effect of Lp(a) in the induction of calcification in vitro on HAVICs and an abnormal accumulation of apolipoproteins and phospholipids in diseased valves [30]. Given all this evidence on oxidized lipids and AS, the recent sub-study of the ASTRONOMER randomized clinical trial failed to find an association between autoantibody titers (which is an indirect measurement oxidation-specific epitopes) and progression of aortic stenosis or the need for AVR [31]. Another very recent study on CAVS samples revealed elevated Lp(a) and OxPL levels thanks to 18F-sodium fluoride positron emission tomography/computed tomography and, above all, found aortic valve microcalcification before the development of clinical manifestation of the pathology [32]. Firstly, this evidence is important because of the implication of Lp(a)/OxPL CAVS disease initiation, and secondly, because the imaging tool used is noninvasive and already used in clinical trials.

Another in vitro study showed that VICs participate in stenosis development by differentiating into the pathological phenotype in response to ECM stiffness [33]. Additionally, the inflammatory condition leads to the differentiation of a subpopulation of VICs into myofibroblasts, which are responsible for the aggregation of fibrous tissue and remodeling of the ECM [34]. Increased inflammation, activation of leukocytes through the secretion of tumor necrosis factor α (TNF-α), and macrophages, dendritic cells, T cells, B cells and mast cells stimulate these valvular myofibroblasts to proliferate and express MMPs [35,36]. The upregulation of MMPs and other proteolytic enzymes is a feature of diseased aortic valve cusps: conditions of continuous static pressure result in higher collagen production in porcine aortic valve leaflets and, in addition, biosynthetic function of these leaflets are influenced by cyclic pressure in a magnitude- and frequency-dependent way [37]. Thus, excessive mechanical stretch on aortic valve leaflets can be detrimental to proteolytic enzyme expression and activity and accelerates disease progression [38].

Another feature regarding the stenosis is the calcification process: the degree of calcification is associated with the expression of osteopontin, which is a protein colocalized with valvular calcific deposits [39] that modulates calcification in valve tissue [40]. Calcified cardiac valves show heterotopic ossification with the formation of mature lamellar bone, microfractures, and hematopoietic tissue [41]. The link between the calcification process and bone development has also been confirmed by evidence that bone sialoprotein (BSP, a noncollagenous component of bone ECM) and bone morphogenetic protein-2 (BMP-2, a ligand of the transforming growth factor-beta (TGF-β) superfamily of proteins) are differentially expressed in normal aortic valves and aortic stenotic valves [42]. In the latter, the gene expression of several osteogenesis pathway players, including low-density lipoprotein receptor-related protein 5 (Lrp5), SRY-box 9 (SOX9), runt-related transcription factor 2 (RUNX2), osteocalcin, BSP, osteopontin and alkaline phosphatase (ALP) [43,44], is induced in a process comparable to skeletal bone development controlled by an osteoblast-like phenotype [44]. 

The Notch homolog 1 translocation-associated (NOTCH1) pathway is also involved in aortic valve calcification. NOTCH1 normally represses RUNX2 function as a fundamental transcriptional regulator of osteoblast lineage; when mutated, NOTCH1 causes an early developmental impairment in the aortic valve and severe valve calcific process later [45,46,47]. In fact, areas of calcification in human aortic valves presented low levels of NOTCH1, and in vitro studies inhibiting NOTCH1 signaling stimulate aortic valve calcification via regulation of SOX9 [48]. Furthermore, recently, it has been found that a downregulated expression of NOTCH1 in CAVS tissue is linked to an augmented methylation level of its promoter; this condition in HAVICs leads to Wnt/β-catenin pathway activation and to osteogenic differentiation [49].

Following mechanical strain, VICs start to produce and accumulate spheroid-mineralized microparticles through a mechanism that is dependent on Ras homolog family member A/Rho-associated protein kinase (RhoA/ROCK) and the transport of ectonucleotide phosphodiesterase/pyrophosphatase-1 (ENPP1) to the plasma membrane [50]. Using nanoanalytical electron microscopy techniques, these spherical calcium phosphate particles have been defined as different from those involving bone composition [51].

Receptor activator of NF-kappa B ligand (RANKL) is an essential cytokine for bone remodeling; it is the ligand for the osteoclastic RAN factor κ (RANK). As long as RANKL interacts with osteoprotegerin (OPG), it cannot activate RANK. Modification of these interactions in vivo prevents both osteoporosis and vascular calcification [52]. In cultured human myofibroblasts, obtained from AV, RANKL stimulates matrix calcification by enhancing the expression of osteoblast-related genes, such as ALP and osteocalcin, confirming the shift to osteogenic program and vascular calcification [53]. In addition to the pro-osteogenic shift in CAVS, different studies have focused on the important role of extracellular vesicles (EVs) in the genesis of calcific nodules within the valve structure. These mediators derived from SMCs, VICs and macrophages, but the underlying mechanisms require further investigation [54]. For example, VICs release EVs that influence VECs and permit correct cell interaction to maintain valve homeostasis [55].

To better study the physiological status of calcified aortic valves, several 3D bioprinted CAVS models have been taken advantage of (well discussed in [56]). A 3D model with encapsulated HAVICs able to reproduce leaflet-layer biomechanics was created for the first time recently, and it helps in the study of AS in an environment more similar to reality [57].

In addition to these molecular transformations, in this review, we analyzed the role of mitochondria and autophagy in the complex process of aortic calcification.

### 3.1. Mitochondrial Dysfunction

Mitochondria are present in high numbers in cardiac cells due to the extremely high energy demands to support contractile activity of the heart compared to other organs. As a consequence, impaired mitochondrial function might lead to several cardiovascular diseases. Furthermore, mitochondria perform several key functions, including the regulation of apoptosis, reactive oxygen species (ROS) production, mitochondrial permeability transition, and Ca^2+^ homeostasis [10,58,59,60] but also development, immune signaling and autophagy via the transduction of cytosolic signals into precise changes of mitochondrial shape and function [61,62]. The mitochondrial membrane potential (ΔΨm) is the driving force for ATP production during oxidative phosphorylation, but also for the removal of impaired mitochondria (mitophagy) and for the transport of proteins and ions such as Ca^2+^ [63]. The loss of ΔΨm leads to a retrograde response, in which mitochondria regulate several signal transduction proteins to control stress-responsive genes in the nucleus [64]. Mitochondrial coupling is strictly related to ATP production, ROS generation and Ca^2+^ homeostasis, and all these processes are finely tuned to maintain physiological conditions [65,66]. Given their importance, mitochondria have crucial roles in the pathophysiology of several cardiovascular diseases. Failure of the normal processes controlled by mitochondria leads to abnormal ROS production and damage to mtDNA, which is considered one of the molecular mechanism leading to initiation and progression of atherosclerosis [59,67,68]. It is known that CAVS displays some similarities to atherosclerosis, including the same risk factors [69], evidence of endothelial dysfunction [70], impaired oxidative stress [71] and other determinants [72]. Oxidized low-density lipoproteins (OxLDLs) are proapoptotic factors that are present both in CAVS and atherosclerotic plaques, and they control two mitochondrial pathways: the calcium/calpain/mitochondrial permeability transition pore (mPTP)/cytochrome C/caspase-3 axis and the release of apoptosis-inducing factors due to calcium dysregulation [71]. These apoptotic pathways might also be associated with AS pathology.

Recent evidence highlights the involvement of mitochondria in the onset and advancement of AS. In 2019, mitochondrial DNA haplogroups were analyzed in a cohort of patients who underwent AVR connected to a critical AS. It was found that these mtDNA sequence polymorphism variations could be linked to the severity of the pathology. Haplogroup H might be considered a risk factor for AS pathology, whereas Uk has a positive role in the protection of the Spanish population [73].

Another essential function of mitochondria is to participate in the unfolded protein response (UPR) to avoid the accumulation of misfolded proteins. This signal transduction pathway induces the transcription of molecular chaperones, proteases, and antioxidant enzymes that are located mainly in the matrix of mitochondria. Very recent work on in vitro cardiomyocytes and in vivo hearts exposed to different stresses have shown a role of mitochondrial (mt) UPR in cardioprotection, in particular, in myocardial tissue and blood from patients with AS, mtUPR showed a cardioprotective effect, in which the activation of this pathway leads to lowered cell death and fibrosis, decreased plasma levels of biomarkers released from heart injury (high-sensitivity troponin T) and functional impairment (N-terminal pro-B-type natriuretic peptide) [74].

Furthermore, cardiac biopsies of ventricular apex and atria of patients with AS showed the presence of a metabolic change from fatty acid oxidation to increased utilization of glucose by a reduction in fatty acid translocase (FAT/CD36) protein, which is the predominant fatty acid transporter in cardiac tissue; in addition, AS biopsies presented an increase in expression of glucose transporter (GLUT) 1 and 4. It has also been revealed that there is downregulation of proteins related to β-oxidation, Krebs cycle and oxidative phosphorylation, suggesting a fundamental correlation of mitochondrial functionality in disease progression, even if further evidence has to be found to define mitochondrial impairments as the cause or the consequence of AS [75].

A recent study found out a new role for dynamin-related protein 1 (DRP1), a key protein that controls mitochondrial fission in calcified human aortic valves where its RNA-mediated interference in HAVICs is able to reduce the calcification process through the increase in SOX9 expression [76].

Vascular endothelial growth factor (VEGF) has revealed a protective effect on heart tissue in an in vivo myocardial infarction model via VEGF-PI3k-AKT through a reduction in apoptotic pathways controlled by mitochondria [77]. These effects were confirmed in an animal model of AS in which VEGF contrasted the transition from compensatory cardiac hypertrophy to heart failure after AS through a programmed cell death controlled by mitochondria [78]. 

Ca^2+^ homeostasis might also be associated with AS, and a genome-wide association study in human aortic valves showed that mRNA expression levels of the *RUNX2* gene were upregulated in calcified valves. In addition, the calcium voltage-gated channel subunit alpha 1 C (*CACNA1C*) gene, which encodes a subunit of a voltage-dependent calcium channel [79], was found as well upregulated in stenotic valves, implying a link between Ca^2+^ signaling pathway and AS development [80].

### 3.2. Mitochondria and Inflammation

Inflammation is the best-studied process among those related to valve calcification. Overall, inflammatory pathways are enhanced in several cardiovascular diseases, and linked with their development and progression, as reviewed in [81,82,83]. Mitochondria are organelles that decode signals regarding the proinflammatory status and regulate immune cells during infection and tissue injury [84]. These organelles act on innate immunity through redox-sensitive mechanisms and contribute to inflammasomes activation [85,86], which are a number of multiprotein complexes that assemble and stimulate caspase-1 and promote the maturation and release of interleukin (IL) 1β and IL-18 from inactive precursors. In particular, inflammatory conditions are characterized by excessive ROS produced by the mitochondria, leading to cellular damage. These impaired mitochondria are normally removed by mitophagy, a specific form of autophagy [87]. The key role of mitochondrial ROS has been associated with oxidation of mtDNA and activation of the NOD-, LRR- and pyrin domain-containing protein 3 (NLRP3)-inflammasome; both ROS generation and inflammasome activation are suppressed by inhibition of the voltage-dependent anion channel (VDAC), which is crucial for mitochondrial activity [88]. Evidence has confirmed that the mitochondria-dependent apoptotic pathway is an important step in NLRP3-inflammasome activation: mitochondrial impairment that causes apoptosis is linked to NLRP3 activation and IL-1β secretion, which was confirmed through inhibition of apoptosis by overexpressing B cell lymphoma 2 (Bcl-2), resulting in attenuated IL-1β release [89]. However, the innate immune response is also regulated by auto/mitophagic processes, and the depletion of microtubule-associated protein 1 light chain 3 (LC3)-II and Beclin 1 proteins promotes the activation of caspase-1, production of IL-1β and IL-18 following lipopolysaccharide (LPS) and ATP stimulation. This enhanced NLRP3-dependent inflammation provokes the increase in impaired mitochondria and is characterized by an aberrant ROS generation and the release of mtDNA into the cytosolic compartment [90].

The role of inflammation in the degeneration of the aortic valve is essential in understanding the calcification process; thus, establishing which immune cells are involved in AS might help to develop efficient therapeutic approaches. In addition, the previously described mitochondrial impairment is a feature of this pathology and is strongly associated with the inflammatory state that is typical of AS samples. Increased levels of inflammation in patients with AS have been confirmed by positron emission tomography (PET), revealing important correlations with disease severity [70], and several immune cells, such as macrophages and a few T cells, have been found to localize at the site of early lesions of degenerative AS [19]. Macrophages are activated in valve tissue and initiate the inflammatory process, releasing cytokines, such as IL-6, which results in high expression in mineralized aortic valves. Through a transcriptomic approach, a novel mechanism has been identified in VICs whereby IL-6 function is controlled by P2Y purinoceptor 2 (P2Y2R) and AKT, which are able to modulate the nuclear factor kappa-light-chain-enhancer of the activated B cell (NF-κB) pathway, promoting the mineralization process [91]. The macrophage phenotype has been separated into classically activated M1 macrophages (usually activated by interferon gamma (IFN-γ) and lipopolysaccharide (LPS) to initiate an immune response) and alternatively activated M2 macrophages (activated by other cytokines associated with wound healing and tissue repair). The increased infiltration of M1 macrophages is significantly higher in calcific aortic valve samples than in control counterparts, and the secretion of TNF-α and IL-6 promotes osteogenic differentiation [92]. In this case, the number and type of macrophages classify the M1 polarization status as a potential calcification inducer; indeed, these cells seem to stimulate aortic valve calcification through macrophage-derived microvesicles (MVs) that contain miR-214, which targets Twist family BHLH transcription factor 1 (TWIST1) in VICs, downregulating its expression [93]. Macrophages also secrete TNF-α, which is a pleiotropic cytokine; treatment of HAVICs with TNF-α, significantly increases the gene expression of calcification markers, such as *ALP, BMP-2*, and distal-less homeobox 5 (*Dlx5*), accelerating the calcification of HAVICs compared to that of the control group [94]. Of note, in vitro administration of TNF-α to AV myofibroblasts induced an increased development of calcified, ALP-enriched cell nodules and concentration of osteocalcin, generating an osteoblast-like cellular phenotype [95]. 

As previously mentioned, investigation of human calcified areas in aortic valves has evidenced significant leukocyte infiltration and the presence of IL-1β and MMP-1. The role of IL-1β has been linked to remodeling of the ECM through the control of MMP expression [96] in human atherosclerotic lesions; experiments on human SMCs treated with IL-1 or TNF-α showed increased synthesis of MMPs, which was consistent with the results of immunohistochemical studies on human arteries and carotid atheroma [97]. IL-1β and TNF-α increased total matrix MMP activity in vitro in cultures of cardiac rat fibroblasts, leading to alterations in the ECM [98]. 

Additionally, activated CD8^+^ T lymphocytes are present and release high level of the inflammatory cytokine IFN-γ, preventing calcium resorption by valve osteoclasts and leading to the initiation of calcification [99]. 

Within the ECM of calcified valves, high levels of TGF-β1 were also detected in studies on sheep AVICs, and this cytokine is involved in the formation of ALP-enriched apoptotic nodules and consequent calcification [100]. 

Most likely, the real pattern of inflammatory cytokines is not completely understood, even if a 2012 report characterized cellular infiltrates in CAVS and revealed the presence of CD4^+^ T lymphocytes, CD20^+^ B-lymphocytes, polyclonal plasma cells, histiocytes and mast cells [101]. In particular, CAVS contain expanded T cell clones with variable degrees of polyclonality, and this is directly proportional to calcification levels [102]. 

The presence of B cells in valve tissue is linked to worsening of the stenosis: these cells are activated by macrophage release of B cell-activating factor (BAFF), a member of the TNF family where the degree of calcification is linked to BAFF receptor expression [103]. 

Mast cells also participate in AS progression; in particular, in addition to the increase in number in CAVS compared to that of nonpathological conditions [104], when activated, mast cell production of cathepsin G leads to adverse valve remodeling and elastin fragmentation [105]. Furthermore, these cells have an essential contribution in regulating angiogenesis during AS progression by producing VEGF to promote vascularization and tryptases to degrade the antiangiogenic molecule endostatin, accelerating valvular neovascularization [106]. 

Inflammation is supported by angiogenesis in the valve environment; increased inflammation occurs close to calcification sites, where neovessels with augmented expression of the intercellular adhesion molecule-1 (ICAM-1) and vascular cell adhesion molecule-1 (VCAM-1) are found, suggesting an active immune-mediated process [107]. Another study on aortic valves from patients confirmed that pathological ones presented a high number of soluble adhesion molecule levels, such as ICAM-1, VCAM-1 and endothelial selectin (E-selectin), which are manifestations of inflammatory conditions [108]. 

To further demonstrate the involvement of inflammation in AS progression, a gene expression study with microarray experiments performed on five AS valves and five normal aortic valves revealed upregulation of several chemokines and relative receptors in aortic stenotic valves compared to that of their control, in addition to ECM remodeling enzymes [109]. In a subsequent study, microRNA (miRNA) features of stenotic AV were investigated, and an increased expression of genes implicated an inflammatory and immune response, such as C-C motif chemokine ligand 3 (*CCL3*) and *CCL4* (which is particularly targeted by miR-125b in macrophages) chemokines was observed [110].

### 3.3. Mitochondria and Oxidative Stress

Mitochondrial dysfunction and subsequent dysregulated ROS generation lead inevitably to extensive oxidative stress, DNA damage, peroxidation of proteins and lipids, and activation of mitochondrial-driven cell death (i.e., where the opening of mPTP is involved) [111]. These cellular impairments are found in many cardiac diseases, including AS; in fact, diseased aortic valves exhibit an increased level of oxLDL. For 20 years, high levels of oxLDL have been associated with apoptosis due to cytochrome c release into the cytosol and trigger of the caspase cascade [112,113]. In addition, as mentioned above, oxLDL is capable of activating the innate immune response, promoting the infiltration of immune cells into the tissue [114]; in fact, an association was found between increased levels of circulating oxLDL and higher fibrosis and calcification in the aortic valve of AS patients [115].

A hypothesis to explain the physiological transition in disease progression is that myofibroblasts in aortic valves differentiate into osteoblasts, as it has been demonstrated that Lrp5, osteocalcin, and other osteochondrogenic differentiation markers were amplified in calcified aortic valve [43]. In vitro, both minimally oxidized low-density lipoprotein (MM-LDL) and lipid oxidation metabolites inhibited proliferation, dose-dependently increased ALP function (which is a marker of the osteoblastic phenotype), and formed extensive areas of calcification in calcifying vascular cells (CVCs) [116].

Among the mitochondrial electron transport chain (mETC) complexes, I and III are recognized as the major sources of superoxide [117,118], which is dismutated by the superoxide dismutases (SODs) to H_2_O_2_ [119]. It has been demonstrated that calcified regions of stenotic valves have significantly reduced mRNA levels of copper-zinc-superoxide dismutase (CuZnSOD), manganese-dependent superoxide dismutase (MnSOD), and extracellular superoxide dismutase (ecSOD) compared with those of normal tissue and a consequent increase in superoxide in these regions [120]. The same findings were found in hypercholesterolemic mouse models, which are prone to calcification and are characterized by increased oxidative stress at the aortic valve level [121]. This was further confirmed by experiments on both a rabbit model of aortic valve calcification and in specimens from humans with aortic valve sclerosis, providing evidence that ROS, particularly H_2_O_2_, potentiate aortic valve calcification; indeed, lipoic acid treatment, which facilitates H_2_O_2_ metabolism, decreases calcification in vivo [122].

### 3.4. Nitric Oxide Synthase

Endothelium-derived nitric oxide (NO) is produced by endothelial nitric oxide synthase (eNOS) and it is a fundamental regulator of apoptosis and cell growth, in addition to acting as a vasodilator with antithrombotic properties [123]. Thanks to in vitro and in vivo studies, an additional role for endothelium-derived NO as a signal from VECs to VICs in preventing calcification through the modulation of the NOTCH1 pathway has been discovered. [124]. In a very recent publication, the effects of L-Arginine, the main precursor of NO, on bovine aortic interstitial valve cells and on HAVICs have been investigated. The results revealed a reduction in matrix calcification due to reduced ALP activity and calcium deposition. Besides, L-Arginine is able to reduce the inflammatory activation of cells [125]. It is known that the uncoupling of NOS is involved in the generation of superoxide; increased protein expression and functional activity of eNOS in rabbit aortic valves upon treatment with atorvastatin leads to diminished calcification and reduced oxidative stress and inflammation [126]. As a confirmation of this, in vitro experiments on calcifying nodules induced by TGF-β1 in porcine aortic VICs showed that NO release inhibits calcification both via increased intracellular cyclic guanosine monophosphate (cGMP) and reduced intracellular superoxide concentrations [127]. The fundamental role of NO was confirmed both in HAVICs and in vivo mouse models. NO depletion in VECs activates NF-κB in HAVICs, promoting dipeptidyl peptidase-4 (DPP4) expression and osteogenic differentiation. In addition, the treatment of animal models with sitagliptin, a selective DPP4 inhibitor, led to significant improvements in terms of calcification [128]. Lately, the positive effect on calcification derived from DPP4 inhibition has been linked in vitro to the potentiation of insulin-like growth factor-1 signaling [129].

A new integrated bioinformatics analysis from human AV samples and from human aortic valve fibrosa-derived endothelial cells has predicted different key genes implicated in the initiation and progression of CAVS, among them *NOS3* has been found down-regulated after in vitro mechanical stress [130], and this impairment leads to eNOS uncoupling, increased oxidative stress and consequential endothelium damage [120]. In vivo studies on eNOS-deficient mice have shown a high incidence of BAVs [131], suggesting a connection between eNOS expression and valve development. In fact, studies in vivo on eNOS ^−/−^ mice have shown an increase in fibrotic process of the aortic valve and, in particular, BAVs are more predisposed to calcification even without other identifiable risk factors as compared to trileaflet aortic valves [132]. These pieces of evidence were confirmed in the human tissue of the aortic wall, confirming a strong correlation between eNOS protein levels and aortic valve anatomy in patients with BAVs [133].

### 3.5. Autophagy

Autophagy is a strongly conserved process that is fundamental for homeostatic recovery and degradation of misfolded proteins and impaired organelles. These cytoplasmic constituents are entrapped in double-membrane vesicles and destroyed upon fusion with lysosomes [134,135]. In particular, mitophagy is a type of autophagy that eliminates impaired mitochondria. Briefly, nutrient depletion leads to the inhibition of mammalian target of rapamycin (mTOR) and the subsequent activation of Unc-51-like kinase (ULK1). These events permit the formation of two complexes, autophagy-related 1 (Atg1)/ULK1 and class III phosphatidylinositol 3-kinase (PI3K III)/Beclin1, resulting in initiation/nucleation. The phagophore structure is created from plasma or organelle membranes and becomes the autophagosome when it completely surrounds cytoplasmic materials. These steps of elongation and maturation are controlled by two ubiquitin-like systems, the first involving E1-like enzyme Atg7- and the E2-like enzyme Atg10-dependent conjugation of Atg12 to Atg5 and the second involving the conjugation of phosphatidylethanolamine (PE) to LC3/Atg8 by the consecutive intervention of Atg proteins (4, 7 and 3). At this point, LC3-I is converted to the autophagosome-associated form LC3-II. Finally, the autophagosome moves along microtubules until it fuses with a late endosome or lysosome to produce the autolysosome, and the final step is the maturation or degradation stage [136].

Autophagy and mitophagy are fundamental processes at the cardiac level to remove dysfunctional mitochondria due to aging and accumulation of misfolded proteins [137]. Autophagy and mitophagy impairment have been associated with the pathogenesis of several cardiovascular diseases [87], such as ischemic heart disease [138], cardiac hypertrophy [139], heart failure [140], atherosclerosis [141] and dilated cardiomyopathy [142].

The first paper linking autophagy to the calcification of aortic leaflets is dated 2006; here, immunohistochemical procedures to detect ubiquitin in 24 human histological samples (12 controls and 12 severe CAVS) highlighted the presence of autophagy in pathological conditions and its absence in the control counterparts. Since the apoptotic pathway has been measured but not detected, the authors claimed that autophagy was responsible for the calcification process and, thus, indirectly for cell death [143]. This conclusion needs to be confirmed by additional specific techniques and in a larger cohort of patients.

Conversely, in 2017, a study performed in HAVICs from normal donors and donors with calcification reported significantly higher autophagic activity in healthy leaflets than in CAVS. Modulation of autophagy by the use of the inhibitor 3-methyladenine (3-MA) and the activator rapamycin highlighted the protective role of autophagy in inhibiting BMP and ALP expression and calcium deposits. Thus, autophagy inhibits pro-osteogenic activity in HAVICs [144]. This work shows discrepancies with that of Somers et al. [143]. This could be related to the use of different markers of autophagy and ubiquitin detection in the whole tissue versus LC3 evaluation in living cells.

Carracedo et al. recently found upregulation of the autophagic process exclusively in the calcified valve tissue, following an increase in the protein expression of LC3-II and ATG and the downregulation of ULK1, confirming the preliminary results from 2006 [145]. The explanation of the process was different from previous reports; here, autophagy acts as a survival mechanism in HAVICs and not as a cell death mechanism substitution for apoptosis. This would guarantee protection from calcification.

Moreover, the autophagic process in HAVICs seems to be influenced by the concentration of inorganic phosphate (Pi), which can control cell survival as a specific cell sensor. Low/moderate levels of Pi lead to atypical autophagy and act as a cell protection mechanism against death and mineralization, while high concentrations of Pi trigger calcification-mediated cell degeneration [146].

In conclusion, no definitive results are available. Each study reported different findings with equally different explanations (Figure 3). However, autophagy is positively or negatively associated with the calcification of aortic leaflets, and further studies are needed to address its significance.

## 4. Efforts to Find Good Therapeutic Strategies

Since an effective therapeutic approach for AS is still lacking, it might be interesting to develop possible novel medical strategies to study the similarities between the pathogenesis of AS and several other pathologies.

For example, atherosclerosis presents many common risk factors, such as endothelial damage, lipid deposition, angioneogenesis and inflammation, similar to those of AS [13]. In a retrospective study in patients with 3-hydroxy-3-methylglutaryl-coenzyme A (HMG-CoA) reductase inhibitor administration, slowed progression of aortic valve disease and a significant reduction in aortic valve calcium (AVC) accumulation were identified, suggesting that this treatment might be useful to modulate this signaling pathway to target early stages of the pathology [147]. Recently, in 2008, a prospective, randomized, placebo-controlled study denied previous findings, demonstrating the lack of effect of an HMG-CoA reductase inhibitor on patients with asymptomatic CAVS [148]. In fact, several prospective randomized controlled trials failed to prove the positive effect of statin therapy on AS progression and clinical outcomes [149,150,151], suggesting an important difference in the two pathological processes. Another important disease is hypertension, which is featured by a significant higher pressure afterload and hypertrophy of the left ventricular chamber which is controlled by the renin–angiotensin–aldosterone system; these negative effects on heart functionality are very similar to AS. A retrospective study in 2005 associated the use of angiotensin-converting enzyme (ACE) inhibitors with a reduced rate of AVC accumulation [152], but a previous study had just the opposite results, demonstrating that ACE inhibitors did not show an effect in slowing AS progression [153]. Osteoporosis patients display an increased incidence of AS with rapid progression [154]. Vascular calcification is frequently combined with reduced bone mineral density or disturbed bone turnover, and this event is called the ’calcification paradox’ [155]. In particular, since 1998, in vivo studies on OPG-deficient mice have demonstrated calcification of the aorta and renal arteries in addition to osteoporosis development, suggesting an important involvement of this pathway in both pathologies [156]. The assumption that artery calcification is associated to bone resorption was confirmed thanks to evidence derived from in vivo studies showing that bisphosphonate administration leads to a reduction in the calcification of arteries and heart valves [157]. Furthermore, bisphosphonates are associated with bioprosthetic porcine aortic valve cusps that showed calcification resistance in long-term implant studies in vivo [158].

It is not entirely known whether AS is a result of an inflammatory pathology or if inflammation is a connected mechanism that occurs subsequently in damaged tissue and in repair mechanisms. However, if and how anti-inflammatory drugs can diminish and block the level of calcification and osteogenesis generated by inflammatory mediators remains an important question to be answered. Utilizing calcinosis-reducing drugs (such as bisphosphonate, denosumab, ectonucleotidase, and ACE inhibitors) for AS therapy focusing on the calcinosis step has been proposed [159,160].

At present, interest is also focused on proprotein convertase subtilisin/kexin type 9 (PCSK9) inhibitors, which are medications able to regulate both lipoprotein (a) and dyslipidemia, and may have a potential in controlling the progression of AS.

Some evidence also derived from the treatment of an in vivo model with the ectonucleotidase inhibitor ARL67156 (6-N,N-diethyl-D-β,γ-dibromomethylene ATP trisodium salt). Ectonucleotidases metabolize nucleotides into phosphate products and have been shown to influence calcification of the aortic valve; in fact, in a rat model, the inhibition of ectonucleotidase activity inhibited the progression of calcification in the aortic valve [161]. Therefore, ectonucleotidases might be a putative target in the treatment of CAVS. 

Despite all these promising areas of research, today, the only effective available treatment for AS is surgical or transcatheter AVR with prostheses when the subject is symptomatic and the valve area is severely reduced. No medical therapies are available to delay disease progression.

## 5. Conclusions

Despite the involvement of mitochondrial dysfunction in almost all cardiovascular diseases, its involvement in AS is widely demonstrated by the inflammation process and related oxidative stress. This involvement is speculated to be the result of the activation of other mechanisms, such as those referring to calcium handling and dynamics. The implication of autophagy in AS, although with few and very conflicting results, provided the possibility to deepen our knowledge on the molecular pathways activated in severely calcified valves and the putative involvement of mitophagy as a mechanism switched on for either basal or excessive removal of impaired mitochondria during cell phenotype remodeling [87].

## Figures and Tables

**Figure 1 ijms-21-04899-f001:**
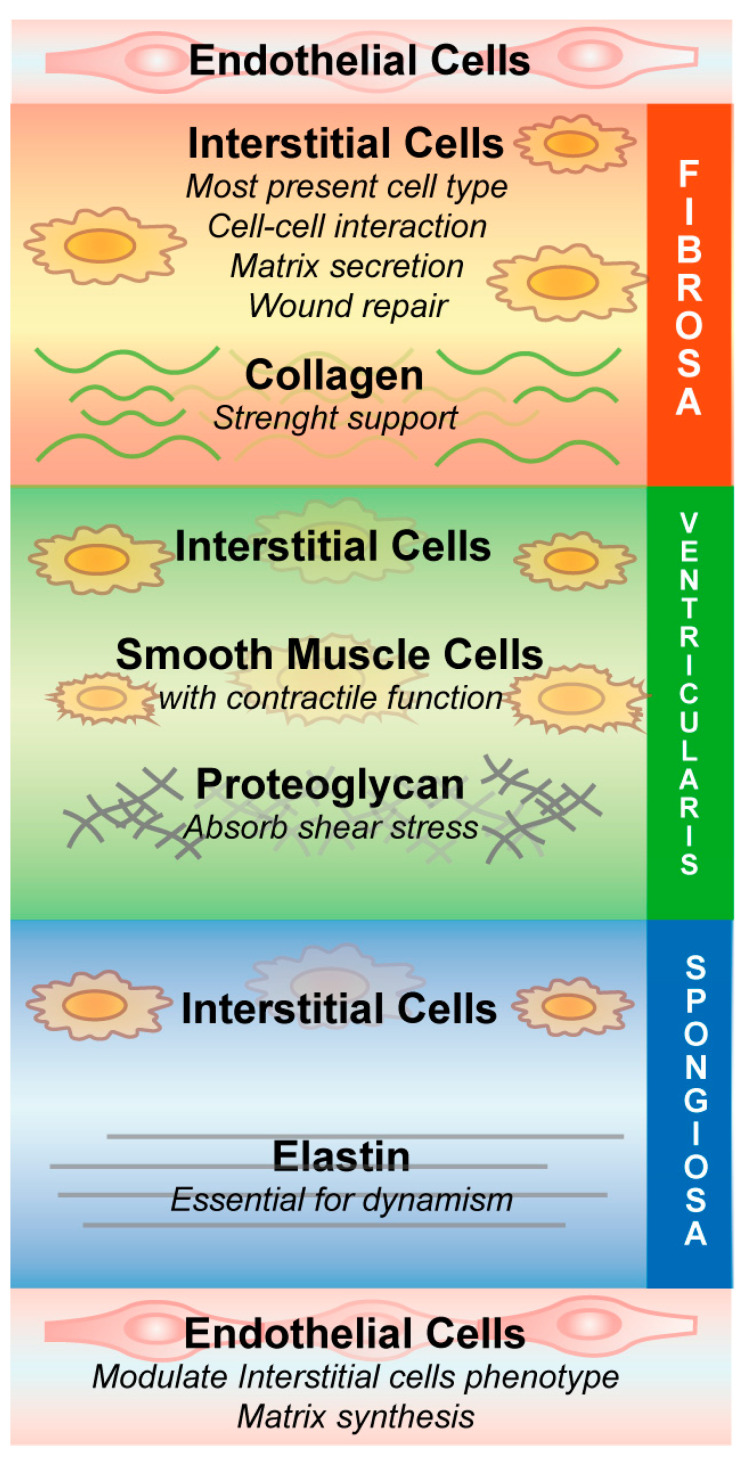
Aortic valve structure. Aortic valve is divided into three different layers: the fibrosa which faces the aorta, the spongiosa in the center and the ventricularis which faces left ventricular outflow tract. Valvular interstitial cells are the prevalent cell type and present in all layers. In addition, the tissue is characterized by a small percentage of smooth muscle cells and by valvular endothelial cells. All these cells are surrounded by extracellular matrix, including collagen, proteoglycan and elastin.

**Figure 2 ijms-21-04899-f002:**
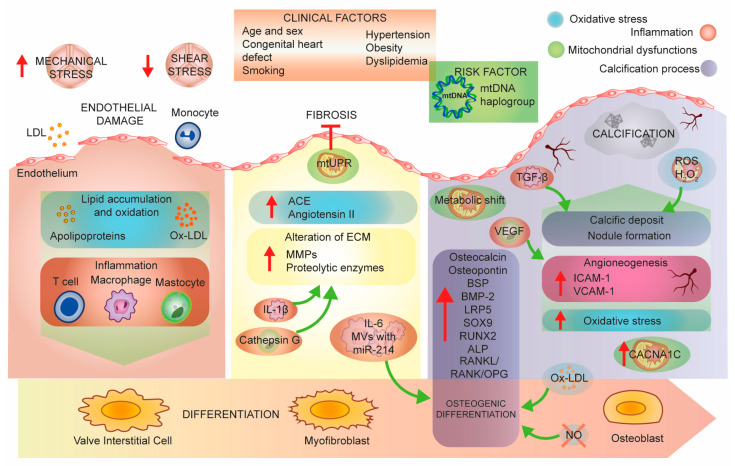
Aortic stenosis pathobiology: a molecular perspective. Mechanical and shear stress induces endothelial damage that leads to lipid accumulation and increased inflammation in valve tissue. These physiological impairments lead to fibrotic processes and to a step-by-step differentiation of valve interstitial cells to myofibroblasts and finally osteoblasts. The final stage of the disease is the calcification of the tissue. Abbreviations: LDL, low-density lipoprotein; oxLDL, oxidized low-density lipoprotein; mtUPR, mitochondrial unfolded protein response; ACE, angiotensin-converting enzyme; ECM, extracellular matrix; MMPs, matrix metalloproteinases; MVs, microvesicles; TNF-α, tumor necrosis factor α; BSP, bone sialoprotein; BMP-2, bone morphogenetic protein-2; LRP5, LDL receptor-related protein 5; SOX9, SRY-box 9; RUNX2, runt-related transcription factor 2; ALP, alkaline phosphatase; RANKL, receptor activator of NF-kappa B ligand; RANK, receptor activator of nuclear factor κ; OPG, osteoprotegerin; NO, nitric oxide; VEGF, vascular endothelial growth factor; TGF-β, transforming growth factor beta; ROS, reactive oxygen species; ICAM-1, intercellular adhesion molecule 1; VCAM-1, vascular cell adhesion molecule-1; CACNA1C, calcium voltage-gated channel subunit alpha 1 C.

**Figure 3 ijms-21-04899-f003:**
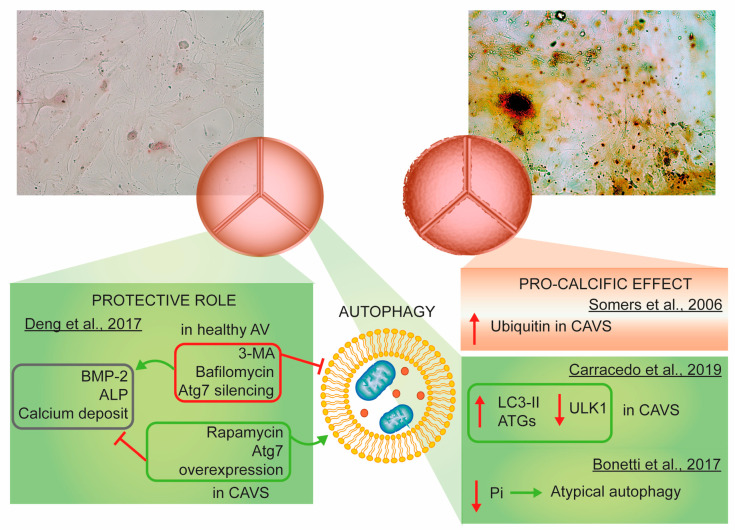
Autophagy involvement in aortic stenosis. Immunohistochemical alizarin red staining was used to identify calcium deposits in the human aortic valve interstitial cells. The autophagic process has been associated with the calcification of aortic leaflets by several independent works, but no definitive results have been achieved. Abbreviations: AV, aortic valve; BMP-2, bone morphogenetic protein-2; ALP, alkaline phosphatase; 3-MA, 3-methyladenine; Atg7, autophagy-related 7; CAVS, calcific aortic valves; LC3-II, microtubule-associated protein 1A/1B light chain 3; ULK1, unc-51-like kinase; Pi, inorganic phosphate.

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
