# Peer review of "Aortic Valve Stenosis and Mitochondrial Dysfunctions: Clinical and Molecular Perspectives"

_ijms, 2020, doi:10.3390/ijms21144899_

Round 1
Reviewer 1 Report
The work is worthy and although a review on stenosis has been done recently, their approach differs from this. Nevertheless, this review lacks more in depth mentioning of some issues and although limitation exists on references, the large extension of text without references is not suitable in a review. Moreover, the focus only on mitochondria seems poor.
Other issues:
Line 36: please consider a reference. In fact, Aortic stenosis (AS) is the most prevalent form of cardiovascular disease in the Western world after hypertension and coronary artery disease.
The reference Global, Regional, and National Burden of Calcific Aortic Valve and Degenerative Mitral Valve Diseases, 1990–2017 https://www.ahajournals.org/doi/10.1161/CIRCULATIONAHA.119.043391
Needs to be placed and discussed
Line 97 -107 need references. And so on.
The introduction should be reduced and placed by a figure of the structure. Please consider these changes, please. It will make it more appealing.
Although mitochondria is key on cardiac homeostasis, this reviewer fails to understand lines 191-212 that mention cardiomyocytes mitochondria and not aortic stenosis or the cells of the valves. It is not directed. Moreover, 237-238, do not authors consider that those changes are a result of the stenosis and not a beginning of pathology?
The colors of figure 1 and figure 2 are not clear and pretty and the figure should be enlarged and better quality.
In terms of format, the authors exaggerate on doing paragraphs even when the ideas mentioned are linked with previous paragraph.
Nitric oxide synthase (NOS) should have its on chapter and not placed in mitochondria and inflammation heading.
Line 466 and others, osteoporosis is not a clinical perspective but the authors make a different paragraph on the pharmacological aspects: do not make paragraph. Sometimes the authors lack in objectivity.
I disagree with authors that only focus on mitochondria and strongly overlook other mechanisms. The authors should refresh the data of a 10 year old revision: https://pubmed.ncbi.nlm.nih.gov/19386374/
(Molecular and Cellular Mechanisms of Aortic Stenosis)
Author Response
Reviewer 1
The work is worthy and although a review on stenosis has been done recently, their approach differs from this. Nevertheless, this review lacks more in depth mentioning of some issues and although limitation exists on references, the large extension of text without references is not suitable in a review. Moreover, the focus only on mitochondria seems poor.
We thank the reviewer for comments about this review. We tried to address all the following concerns as indicated. About limitation on references, we added many others in the missing parts and, in general, to update the manuscript with the latest works on the topic. Unfortunately, little is known about the topic; indeed, our aim was to summarize direct and indirect evidence on mitochondrial contribution in aortic stenosis which could be useful to start new lines of research.
Other issues:
Q.1. Line 36: please consider a reference. In fact, Aortic stenosis (AS) is the most prevalent form of cardiovascular disease in the Western world after hypertension and coronary artery disease.
The reference Global, Regional, and National Burden of Calcific Aortic Valve and Degenerative Mitral Valve Diseases, 1990–2017 https://www.ahajournals.org/doi/10.1161/CIRCULATIONAHA.119.043391
Needs to be placed and discussed
A.1. The reviewer is right, this sentence needs a reference, and we promptly provided to insert the suggested one, a long-term and comprehensive study by Yagdir et al.
Please, see lines 36-39.
Q.2. Line 97 -107 need references. And so on.
A.2. The reviewer is right. We added references in this part of the text and throughout the review when considered necessary.
Q.3. The introduction should be reduced and placed by a figure of the structure. Please consider these changes, please. It will make it more appealing.
A.3. We thank the reviewer for the suggestion; we provided a new figure representing the introduction section that has been reduced.
Please, see the new figure 1 and the new Introduction section.
Q.4. Although mitochondria are key on cardiac homeostasis, this reviewer fails to understand lines 191-212 that mention cardiomyocytes mitochondria and not aortic stenosis or the cells of the valves. It is not directed. Moreover, 237-238, do not authors consider that those changes are a result of the stenosis and not a beginning of pathology?
A.4. We appreciated this comment; thus, we modified the first part of the chapter “Mitochondrial dysfunction.” Since a study describing mitochondrial peculiarities in stenotic aortic valves does not exist, we made a general description of the role of mitochondria in the cardiovascular setting (not only aimed at cardiomyocytes). Furthermore, we specified that there are not enough evidences to demonstrate if mitochondrial functional changes found in AS biopsies are the cause or the consequence of the pathology.
Please, refer to lines 244-278.
Q.5. The colors of figure 1 and figure 2 are not clear and pretty and the figure should be enlarged and better quality.
A.5. We thank the reviewer for its recommendation. We provided a new format of figures with better quality (RGB color and at least 300 dpi). Since we have to upload them as zipped files, we hope this procedure does not affect the final figure version.
Q.6. In terms of format, the authors exaggerate on doing paragraphs even when the ideas mentioned are linked with previous paragraph.
A.6. We provided a new text style to achieve continuity in the reading of the review.
Q.7. Nitric oxide synthase (NOS) should have its on chapter and not placed in mitochondria and inflammation heading.
A.7. We thank the reviewer for the suggestion. We appreciated a lot and provided a chapter focused on NOS related to the topic of the whole review.
Please, see the new 3.4 chapter, lines 402-432.
Q.8. Line 466 and others, osteoporosis is not a clinical perspective but the authors make a different paragraph on the pharmacological aspects: do not make paragraph. Sometimes the authors lack in objectivity.
A.8. We share the reviewer’s comment. We aimed to give the idea that no defined and supported pharmacological strategies exist at the moment. Accordingly, we changed the title of this chapter because of the discrepancy of the subjects discussed.
Q.9. I disagree with authors that only focus on mitochondria and strongly overlook other mechanisms. The authors should refresh the data of a 10 year old revision: https://pubmed.ncbi.nlm.nih.gov/19386374/ (Molecular and Cellular Mechanisms of Aortic Stenosis)
A.9. We are sorry that the reviewer does not fully share our proposal idea. As he said previously, other reviews on AS exist, thus we changed the approach to talk about this topic. We are aware that few is known about mitochondrial dysfunction in AS; this would be a stimulus to improve knowledge in the field. However, we discussed other mechanisms considered fundamental for the description of the pathology, such as lipid deposition and genetic reprogramming (please, refer to the updated chapter 3). About the second comment, we added more recent references in all the text.
Reviewer 2 Report
The authors discussed inflammation and oxidative stress which are the main molecular mediators of the evolution of aortic stenosis in patients; these mediators regulate both the degradation and remodeling processes. The other topics were mitochondria and autophagy as contributors to disease. They argued that a better understanding of these cellular impairments might help to develop new ways to treat patients without surgical interventions.
General comments
This is a manuscript addressing an interesting topic “Aortic valve stenosis and mitochondrial dysfunctions: clinical and molecular perspectives”. This is a well-written review, although some concerns need to be addressed (see comments to the authors for details).
Specific comments
- Line 78: CAVS would be spelled out for the first expression.
- Line 228-240: The references were those using heart muscles but not the AV; It would be deleted or the authors would state that these are not reported in the AV tissues.
Author Response
Reviewer 2
The authors discussed inflammation and oxidative stress which are the main molecular mediators of the evolution of aortic stenosis in patients; these mediators regulate both the degradation and remodeling processes. The other topics were mitochondria and autophagy as contributors to disease. They argued that a better understanding of these cellular impairments might help to develop new ways to treat patients without surgical interventions.
We thank the reviewer for his opinion and comments about this review.
General comments
This is a manuscript addressing an interesting topic “Aortic valve stenosis and mitochondrial dysfunctions: clinical and molecular perspectives”. This is a well-written review, although some concerns need to be addressed (see comments to the authors for details).
We thank the reviewer for his valuable comments. We tried to address all the following concerns as indicated.
Specific comments
Q.1. Line 78: CAVS would be spelled out for the first expression.
A.1. Thank you for this comment, we inserted the description of abbreviation.
Q.2. Line 228-240: The references were those using heart muscles but not the AV; It would be deleted or the authors would state that these are not reported in the AV tissues.
A.2. The reviewer is right, we paid more attention in describing the study, specifying that samples are not of AV but from patients with AS.
Please, see lines 252-265.
Round 2
Reviewer 1 Report
The authors made substantial changes to the work, which improve it.
I recommend acceptance.
Best wishes
Vera